# Building Multi-domain Dialog State Trackers from Single-domain Dialogs

**Qi Zhu, Zheng Zhang, Xiaoyan Zhu, Minlie Huang**[*]

The CoAI Group, DCST, Institute for Artificial Intelligence,
State Key Lab of Intelligent Technology and Systems,
Beijing National Research Center for Information Science and Technology,
Tsinghua University, Beijing 100084, China
zhu-q18@mails.tsinghua.edu.cn
{z-zhang,zxy-dcs,aihuang}@tsinghua.edu.cn

## Abstract

Existing multi-domain dialog state tracking (DST) models are developed based on multi-domain dialogs, which require significant manual effort to define domain relations and collect data. This process can be challenging and expensive, particularly when numerous domains are involved. In this paper, we propose a divide-and-conquer (DAC) DST paradigm and a multi-domain dialog synthesis framework, which makes building multi-domain DST models from single-domain dialogs possible. The DAC paradigm segments a multi-domain dialog into multiple single-domain dialogs for DST, which makes models generalize better on dialogs involving unseen domain combinations. The multi-domain dialog synthesis framework merges several potentially related single-domain dialogs into one multi-domain dialog and modifies the dialog to simulate domain relations. The synthesized dialogs can help DST models capture the value transfer between domains. Experiments with three representative DST models on two datasets demonstrate the effectiveness of our proposed DAC paradigm and data synthesis framework.

## 1 Introduction

In recent years, with the emergence of multi-domain dialog datasets (Budzianowski et al., 2018; Rastogi et al., 2020), the focus of task-oriented dialog (TOD) research has shifted from single-domain dialog to multi-domain dialog. In a multi-domain dialog, there may be relations between domains, often exhibited through the transfer of slots' values as illustrated in Table 1. However, as the number of domains increases, manually defining domain relations and gathering multi-domain dialog data becomes increasingly expensive. In this paper, we investigate how to build multi-domain DST models from single-domain dialogs, which may help scale TOD systems to handle numerous domains.

---

[*]Corresponding authors.

| **Bus domain** |
|---|
| **Usr:** I want bus tickets! |
| **Sys:** Where are you going and what time do you plan to leave? |
| **Usr:** I want to leave on the **3rd of March** at 12:45. I want to visit **San Francisco**. |
| ...(few turns later)... |
| **Usr:** Sounds fantastic. |
| **Sys:** Your reservation is confirmed. |
| **State:** [(Bus, departure_date, **3rd of March**), (Bus, to_city, **San Francisco**)...] |

| **Hotel domain** |
|---|
| **Usr:** Could you search for hotels in **that area** as well? |
| **State Update:** [(Hotel, location, **San Francisco**)] |
| **Sys:** Sure. What is your date of checking in? |
| **Usr:** On **the same day**. |
| **State Update:** [(Hotel, check_in_date, **3rd of March**)] |

Table 1: An example of cross-domain value transfer. The values of *location* and *check_in_date* slots in the `Hotel` domain are transferred from the `Bus` domain.

To model the domain relations in multi-domain dialogs, previous works either employ a copy mechanism (Heck et al., 2020; Jiao et al., 2022) or encode a pre-defined schema graph that indicates slot relationships using prior knowledge (Zhu et al., 2020; Chen et al., 2020; Feng et al., 2022). Nevertheless, these models are data-driven, relying on adequate multi-domain dialog data. Another line of research is conversational query rewriting (CoQR), which rewrites a user query to be self-contained and understandable without the dialog context (Su et al., 2019; Rastogi et al., 2019; Quan et al., 2019). However, these methods also require manually collecting in-domain query rewriting data.

When there is no prior knowledge about the data distribution of multi-domain dialog, building DST models faces two main challenges: (1) how to make models generalize well to multi-domain dialogs involving arbitrary domains, (2) how to enable models to capture domain relations. In this work, we propose a divide-and-conquer (DAC) DST paradigm and a multi-domain dialog synthesis framework to address the challenges. The DAC paradigm segment a multi-domain dialog into single-domain dialogs and perform DST for each

domain separately, bridging the data distribution gap between dialogs involving different domains. In DAC, models only need to predict the state of the currently active domain according to the dialog history of that domain and the state of other domains, which greatly saves the computation of reading all dialog history and predicting all domains' states. Our proposed data synthesis framework combines multiple single-domain dialogs into a multi-domain dialog and simulates value transfer between slots of different domains. The framework includes (1) identifying candidate slot pairs for cross-domain value transfer, (2) concatenating single-domain dialogs based on candidate slot pairs, (3) rewriting utterances to simulate cross-domain value transfer, and (4) filtering problematic rewrites with a value tagger. The synthesized data can be used to train DST models directly or train a CoQR model to resolve cross-domain value transfer before DST. Experiments on two datasets with three representative DST models show that our proposed DAC paradigm and data synthesis framework significantly improve model performance on multi-domain dialogs when only single-domain dialogs are available.

In summary, our contributions include:

1. We propose a divide-and-conquer paradigm for DST, which improves model efficiency and generalization on multi-domain dialogs.

2. We propose a data synthesis framework that can generate multi-domain dialogs using only single-domain dialogs, which enables DST models to identify domain relations without annotations.

3. Experiments show that our proposed methods can substantially boost multi-domain DST models using single-domain dialogs, which reduces the need to collect real multi-domain dialogs.

## 2 Related Work

### 2.1 Multi-domain Dialog State Tracking

Dialog State Tracking (DST) is a key module of TOD systems, aiming at capturing the user goal in the form of (domain, slot, value) triplets during a conversation. The introduction of large-scale multi-domain dialog datasets, such as MultiWOZ (Budzianowski et al., 2018) and SGD (Rastogi et al., 2020), has led to the development of various multi-domain DST models. These models employ different techniques for predicting the value of a slot, such as selecting the value from a pre-defined set (Lee et al., 2019; Chen et al., 2020), extracting

the value from the dialog (Zhang et al., 2020; Heck et al., 2020), and directly generating the value (Wu et al., 2019; Hosseini-Asl et al., 2020). Value prediction can be executed at either turn-level, where models accumulate the predictions of each turn to obtain the complete state (Kim et al., 2020; Lin et al., 2020), or dialog-level, where models predict the complete state directly (Hosseini-Asl et al., 2020; Peng et al., 2021). In this paper, we verify the effectiveness of our propose DAC paradigm and multi-domain dialog synthesis framework on both turn-level and dialog-level generative DST models.

In a multi-domain dialog, slots from different domains can be correlated. The slot relationships can be modeled implicitly through self-attention or graph neural networks (Kim et al., 2020; Ye et al., 2021; Lin et al., 2021a), or explicitly through copy mechanism or a schema graph that encodes prior knowledge (Heck et al., 2020; Jiao et al., 2022; Zhu et al., 2020; Chen et al., 2020; Feng et al., 2022). However, all of these methods are driven by multi-domain data. When multi-domain data are unavailable or insufficient, our proposed method provides a valuable solution.

### 2.2 Multi-domain Dialog Synthesis

To reduce the cost of collecting multi-domain dialogs, some works propose synthesizing them from a limited number of human-annotated dialogs (Kim et al., 2021; Mohapatra et al., 2021; Wan et al., 2022) by fine-tuning pre-trained models to generate new dialogs based on randomly sampled user goals. On the other hand, Li et al. (2021) rewrite the user query according to a counterfactual user goal in order to assess the robustness of DST models to unseen slot values and rare slot combinations. Different from these works that assume the possible domain dependencies are known and real multi-domain dialogs are available, we identify potential domain dependencies and build multi-domain DST models using only single-domain dialogs.

## 3 Background

### 3.1 Notation

Let $C_t = [U_1, S_1, ..., U_t]$ denote a dialog until the $t$-th turn, where $U_i$ and $S_i$ are user and system utterance respectively. The dialog state $B_t = \{(d, s, v)_i\}_{i=0}^{N_t}$ consists of $N_t$ (domain, slot, value) triplets expressed in $C_t$, where the value is not empty and there is only one value for each domain-slot combination. Following Lin et al. (2020), we

denote the state update from $B_{t-1}$ to $B_t$ as $L_t$. $L_t$ has a similar format as $B_t$ but may contain slots with empty values, which indicates that values from $B_{t-1}$ are removed in the $t$-th turn.

We denote the active domains of a turn as $D_t$. If $L_t$ is empty, $D_t$ is the same as $D_{t-1}$. Otherwise, $D_t$ consists of the domains in $L_t$. For $t < t'$ where $t'$ is the first turn that $L_{t'}$ is not empty, we set $D_t$ to domains in $L_{t'}$. A turn $t$ is regarded as a **cross-domain** (CD) turn if there exists a triplet $(d, s, v)$ in $L_t$, such that (1) $v$ is not empty, (2) $v$ appears in previous turns instead of the current turn, and (3) the domain $d$ is not in the active domains of the turns $v$ appears. We define such behavior as cross-domain value transfer. When a dialog has cross-domain turns, it is called a cross-domain dialog.

### 3.2 Generative DST Models

In this work, we use **generative DST** models that generate the values of slots, as they are flexible and require minimal annotation. Specifically, we consider three representative models:

- **T5-DST** (Zhu et al., 2022) predicts the serialized state $B_t$ from the entire dialog context $C_t$.
- **MinTL** (Lin et al., 2020) predicts the serialized state update $L_t$ from the partial context $C_{t-w:t}$ and the previous state $B_{t-1}$, where $w$ is the context window size. We set $w$ to two to include two previous turns. During inference, we input predicted previous state $B'_{t-1}$ instead of $B_{t-1}$ and update $B'_{t-1}$ to $B'_t$ using $L'_t$.
- **SDP-DST** (Lee et al., 2021) predicts the value of each slot independently with entire dialog context $C_t$, domain name and description, slot name and description, and example values (if any) as input.

We use T5-Large (Raffel et al., 2020) as the backbone for all models. For MinTL and T5-DST, we use the same state serialization function provided by ConvLab-3 (Zhu et al., 2022).

## 4 Method

Unlike previous works on multi-domain DST, we assume we only have single-domain dialogs with no access to prior knowledge regarding how domains will correlate in a multi-domain dialog. In this setting, DST models face two challenges: (1) how to generalize to dialogs involving multiple domains, (2) how to model the domain dependencies. We propose a divide-and-conquer DST paradigm and a multi-domain dialog synthesis framework to address the two challenges.

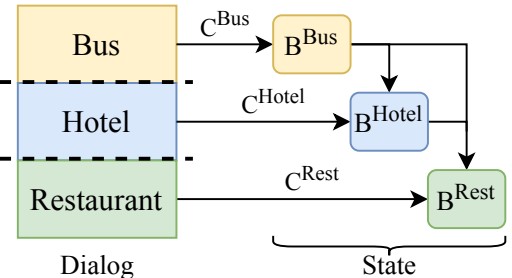

Figure 1: Divide-and-conquer DST paradigm.

### 4.1 Divide-and-Conquer DST Paradigm

As illustrated in Figure 1, in the divide-and-conquer (DAC) DST paradigm, a multi-domain dialog is segmented into single-domain dialogs, then the DST model predicts the state of each domain separately. Formally, for each turn $t$, we identify the active domains $D_t$ using a domain classifier (Dom-CLS). In most turns, $D_t$ contains only one domain. Thus, a multi-domain dialog can be split into several single-domain dialogs. For each $d \in D_t$, the DST model predicts the domain state $B_t^d$ according to domain context $C_t^d$ and previous other domains' state $B_{t-1}^{\backslash d}$, where $C_t^d = C_{t':t}$, $t'$ is the first turn that $d$ appears in $D_{t'}$. By updating previous state $B_{t-1}$ with $\{B_t^d\}_{d \in D_t}$, we get the current state $B_t$.

Compared with existing dialog-level DST models that predict $B_t$ from $C_t$ directly (e.g., T5-DST and SDP-DST), the DAC paradigm has the following advantages:

- The model always takes single-domain dialogs as input, bridging the data distribution gap between dialogs involving varying domain combinations.
- The model takes concise state $B_{t-1}^{\backslash d}$ instead of other domains' dialogs as input, saving the computation and facilitating copying value from other domains.

Besides, the DAC paradigm can greatly improve the efficiency of models that predict the value of each slot independently (e.g., SDP-DST), particularly when there are many domains. In the DAC paradigm, these models only need to predict the state of the active domains instead of all domains.

For turn-level DST models like MinTL that predict state update $L_t$ from partial context $C_{t-w:t}$ with a fixed window size $w$, the DAC paradigm provides a more complete context for domains in $L_t$. According to the definition, for domain $d$ in $L_t$, $C_t^d$ contains all turns that $d$ is active.

We apply the DAC paradigm to T5-DST, MinTL, and SDP-DST models. The domain classifier Dom-

| | |
|---|---|
| $C_t$ | [user] I want bus tickets! ... [user] On the same day. |
| $C_{t-2:t}$ | [user] Sounds fantastic. ... [user] On the same day. |
| $C_t^{Hotel}$ | [user] Could you search for hotels in that area as well? ... [user] On the same day. |
| $D_t$ | [Hotel] |
| $B_{t-1}^{\backslash Hotel}$ | [Bus]([to_city][San Francisco], [departure_date] [3rd of March], ...) |
| $B_{t-1}^{Hotel}$ | [Hotel]([location][San Francisco]) |
| $B_t^{\backslash Hotel}$ | $B_{t-1}^{\backslash Hotel}$ |
| $B_t^{Hotel}$ | [Hotel]([check_in_date][3rd of March], [location][San Francisco]) |
| $B_{t-1}$ | $B_{t-1}^{\backslash Hotel}; B_{t-1}^{Hotel}$ |
| $B_t$ | $B_t^{\backslash Hotel}; B_t^{Hotel}$ |
| $L_t, L_t^{Hotel}$ | [Hotel]([check_in_date][3rd of March]) |
| $X(d_m, s_n)$ | [domain] Name($d_m$) Description($d_m$) [slot] Name($s_n$) Description($s_n$) ([PVs] $v_1, v_2, ...$) |
| T5-DST | $C_t \rightarrow B_t$ |
| DAC | $(B_{t-1}^{\backslash Hotel}; C_t^{Hotel}; D_t) \rightarrow B_t^{Hotel}$ |
| MinTL | $(C_{t-2:t}; B_{t-1}) \rightarrow L_t$ |
| DAC | $(B_{t-1}^{\backslash Hotel}; C_t^{Hotel}; B_{t-1}^{Hotel}; D_t) \rightarrow L_t^{Hotel}$ |
| SDP-DST | $(C_t; X(d_m, s_n)) \rightarrow v_n^m$ |
| DAC | $(B_{t-1}^{\backslash Hotel}; C_t^{Hotel}; X(Hotel, s_n)) \rightarrow v_n^{Hotel}$ |

Table 2: Input and output of DST models with/without DAC paradigm for the example dialog in Table 1.

CLS is a T5-Large model that takes the last two turns $C_{t-2:t}$ as input and predicts active domains $D_t$. Table 2 shows the example input and output. In the DAC paradigm, T5-DST and SDP-DST only need to predict the state of the current domain (Hotel), while dialog contexts of other domains (Bus) are replaced by their states. MinTL with DAC uses the turns of the current domain as input instead of the last two turns. The DAC paradigm could also be applied to other kinds of DST models.

## 4.2 Multi-domain Dialog Synthesis Framework

Previous DST models learn domain dependencies from multi-domain dialogs. Instead of manually collecting multi-domain dialogs, we propose a data synthesis framework to generate multi-domain dialogs from single-domain dialogs automatically. As illustrated in Figure 2, the framework includes the following processes:

1. Mine the slot pairs that could potentially take the same value in a specific dialog context.
2. Concatenate single-domain dialogs that contain relevant slots. Replace target slots' values with source slots' values.
3. Rewrite user queries in the concatenated dialog to implicitly express the value of target slots.
4. Filter out the rewritten queries that miss infor-

mation or contain redundant information.

The synthesized multi-domain dialogs can be used to train DST models directly or to train a CoQR model to rewrite utterances before DST for resolving implicit cross-domain value transfer.

### 4.2.1 Slot Relation Mining

The initial step in synthesizing multi-domain dialogs is identifying the relations between slots from different domains. Given a single-domain dialog of domain $d$, our objective is to identify **target** slots in other domains that can accept the same value as a non-empty **source** slot in $d$. For example, given a dialog in the Bus domain, the *location* slot (target) in the Hotel domain may have the same value as the *to_city* slot (source) in the Bus domain.

Inspired by TransferQA (Lin et al., 2021b), we formulate DST as a Question Answering task to enable zero-shot cross-domain inference, where the input is "what is the {slot description} of the {domain name} domain? {dialog context}" and the output is the slot's value. Initialized by UnifiedQA-v2 (Khashabi et al., 2022), our QADST model is trained on single-domain dialogs. For a dialog of domain $d$, the model is trained to predict the values of slots in $d$. When performing cross-domain inference, the model predicts the values of target slots in other domains. By comparing the predicted target slot's value and the source slot's value, we further filter out slot pairs (source slot, target slot) that are generally less consistent (F1 score lower than 0.1). For each single-domain dialog, we record the target slots predicted to have the same value as a non-empty source slot in the current domain.

### 4.2.2 Concatenation and Value Replacement

We combine multiple single-domain dialogs into a multi-domain dialog. While we illustrate the concatenation of **two** single-domain dialogs for ease of explanation, our method can be readily extended to concatenating **multiple** single-domain dialogs by iterative appending.

We randomly sample the first dialog for concatenation. According to the annotation of target slots in the first dialog, the second dialog is sampled from dialogs with at least one non-empty target slot. To improve the fluency of the concatenated dialog, we remove the last turn of the first dialog if it has an empty $L_t$, as this is typically an indication that the user is ending the conversation. For the second dialog, we substitute the target slots' values in the utterances and states with the correspond-

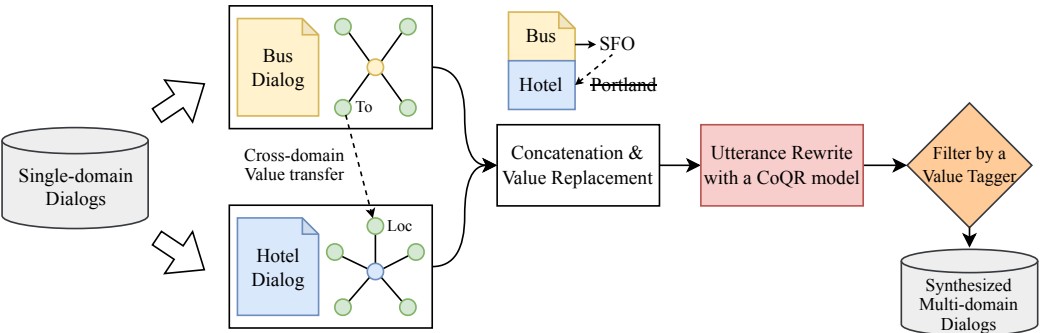

Figure 2: An illustration of proposed data synthesis framework. The graphs near the dialogs are their final dialog state, where the centric nodes are domains and the green nodes are slots. In this example, the value of **Bus**'s *to_city* slot ("SFO") transfers to **Hotel**'s *location* slot. More single-domain dialogs can be merged into a multi-domain dialog by repeating the dialog concatenation, value replacement, and utterance rewriting processes.

---

**Original:** Could you search for hotels in Portland?
**Value Replacement:** Could you search for hotels in **San Francisco**?

**Utterance Rewrite:**
**Input:** Could you search for hotels in `` San Francisco `` ?
**Output:** Could you search for hotels in *that area*?

Table 3: Illustration of value replacement and utterance rewriting.

---

ing source slot values from the first dialog. When appending more dialogs, the subsequent dialogs are processed similarly to the second dialog, except that their target slots can refer to slots of any previous dialogs.

### 4.2.3 Utterance Rewriting

For the turns in which the state update $L_t$ includes target slots for the first time, **the user utterances are further rewritten** to express target slots implicitly, simulating cross-domain value transfer behavior. Instead of manually collecting in-domain query rewriting data for training, we train a CoQR model (CoQR-R) on CANARD (Elgohary et al., 2019) dataset, which contains conversational questions $U^{conv}$ and their self-contained rewrites $U^{sc}$. Unlike Elgohary et al. (2019), our model takes the context $C$ and rewrites $U^{sc}$ as input to predict the original question $U^{conv}$ that may contain anaphora and ellipsis. Since our goal is to express the target slot's value implicitly, we label the modified spans in $U^{sc}$ to make the rewriting more controllable. Specifically, we add special tokens to $U^{sc}$ to indicate the insertion, substitution, and deletion operations needed to convert $U^{sc}$ to $U^{conv}$.

Then the model is used to rewrite the user utterance $U_t^{sc}$ in the second (and subsequent) single-domain dialog to $U_t^{conv}$ when $L_t$ contains target slots. As shown in Table 3, the spans of target slots' values in $U_t$ are labeled with substitution operation tokens. To provide sufficient information for rewriting, the context $C$ consists of the descriptions of the corresponding source slots and their contexts, along with the previous system utterance $S_{t-1}$. Formally, $C = [C_{t_1-1:t_1}, V_{t_1}, C_{t_2-1:t_2}, ...V_{t_N}, S_{t-1}]$, where $t_i$ is the first turn that the $i$-th source slot-value appears in $L_{t_i}$, $C_{t_i-1:t_i} = [U_{t_i-1}, S_{t_i-1}, U_{t_i}]$, and $V_{t_i}$ is constructed by filling a template "The {slot description} is {value}".

### 4.2.4 Filter by a Value Tagger

We further filter the noisy rewritten queries to ensure they do not miss information or contain redundant information. We employ a value tagger to extract the values of the original user query and rewritten one. The rewritten query must not contain the values of target slots and must only contain other values in the original query. The tagger is a RoBERTa-Large (Zhuang et al., 2021) with a token classification head, which assigns one of three tags $\{B, I, O\}$ to each token. It is fine-tuned on the Taskmaster datasets (Byrne et al., 2019, 2021) utilizing dialog acts annotation.

## 5 Experimental Setup

### 5.1 Datasets

Experiments are carried out on two widely used multi-domain TOD datasets, MultiWOZ 2.1 (Eric et al., 2021) and Schema-Guided Dialog dataset (SGD) (Rastogi et al., 2020). We use the datasets processed by ConvLab-3 (Zhu et al., 2022). The SGD dataset covers 45 services from 20 domains, some of which may have multiple services with different schemas. We remove the `Messaging` domain since there are no single-domain dialogs in this do-

| Dataset | Domains (Comb.) | Single-domain Dialogs | Multi-domain Dialogs (CD) |
|---|---|---|---|
| MultiWOZ | 5 (14) | 2824 | 7018 (29.8%) |
| SGD | 19 (42) | 2456 | 3387 (73.8%) |

Table 4: Statistics of the MultiWOZ and SGD datasets.

main. Unlike Rastogi et al. (2020), we assume the services of a dialog are not known beforehand. Therefore, we only retain one randomly chosen service for each domain to avoid confusion. The data statistics are shown in Table 4.

The training, validation, and test sets of the original dataset are merged and randomly re-partitioned to minimize the data distribution gap between training and testing. Single-domain dialogs are divided into training, validation, and test sets in a ratio of 8:1:1. 10% and 50% of the multi-domain dialogs are used for validation and testing, respectively, while the remaining 40% are reserved for experiments that require access to multi-domain dialogs. Domain combinations that have less than 10 dialogs are ignored.

## 5.2 Metrics

The most widely used metric for DST is Joint Goal Accuracy (JGA), which evaluates whether the prediction $B'_t$ and gold state $B_t$ are exactly matched. However, this metric is not smooth. If the model mispredicts even a single turn, this metric will ignore predictions made in all other turns. Therefore, we use Relative Slot Accuracy (**RSA**)(Kim et al., 2022), which can be regarded as a soft version of JGA and can better distinguish models. It is calculated as $\frac{|B_t \cap B'_t|}{|S|}$, where $|B_t \cap B'_t|$ is the number of correctly predicted slots and $|S|$ is the number of unique slots in $B_t \cup B'_t$. We ignore the turns where $|S| = 0$, which means both $B_t$ and $B'_t$ are empty.

We propose a new metric **CDTA** (Cross-domain Turn Accuracy) to measure a model's ability to capture the cross-domain value transfer behavior. CDTA is the Turn Accuracy (TA) of cross-domain turns, examining turn prediction $L'_t$ that updates $B'_{t-1}$ to $B'_t$. Following Dey et al. (2022), instead of comparing $L'_t$ and $L_t$, we regard a turn $t$ as correct when $L'_t \subseteq B_t$ and $L_t \subseteq B'_t$, which credit models for correcting error predictions in previous turns.

## 5.3 Baselines and Training Details

To verify the effectiveness of our proposed data synthesis framework, we compare DST models trained on different data:

| Model | Epoch | Batch | Initialization | Training set |
|---|---|---|---|---|
| QADST | 3 | 512 | UnifiedQA-v2 | SINGLE |
| CoQR-R | 1 | 512 | T5-Large | CANARD |
| Value Tagger | 1 | 128 | RoBERTa-Large | TaskMaster |
| CoQR-Zero | 1 | 512 | T5-Large | CANARD |
| CoQR-SYN | 3 | 512 | CoQR-Zero | SYN |
| DomCLS | 3 | 128 | T5-Large | * |
| DST | 5 | 128 | T5-Large | * |

Table 5: Training settings.

- **SINGLE:** Single-domain dialogs.
- **SINGLE+CONCAT:** Randomly select and concatenate two single-domain dialogs of different domains (CONCAT). Similar to our method, the ending turn of the first dialog is removed. Training models on SINGLE and CONCAT jointly.
- **SINGLE+SYN:** Training models on SINGLE and our synthesized multi-domain dialogs jointly.

Unless otherwise specified, CONCAT and SYN have the same number of dialogs as SINGLE.

We also explore generating self-contained rewrites $U^{sc}$ for **all** user utterances $U^{conv}$ in a dialog before DST during testing. The DST model is trained on **SINGLE+CONCAT** data that do not contain cross-domain value transfer. We consider the following CoQR models to generate rewrite $U^{sc}_t$ given the context $C_t$ and user utterance $U^{conv}_t$:

- **CoQR-Zero:** A CoQR model trained on CANARD dataset (Elgohary et al., 2019).
- **CoQR-SYN:** Fine-tune CoQR-Zero on our SYN dialogs that contain original single-domain utterances ($U^{sc}_t$) and rewritten utterances ($U^{conv}_t$).
- **CoQR-ChatGPT:** Prompting ChatGPT to generate self-contained rewrites (see Appendix C).

**Hyper-parameters** The training details of models are summarized in Table 5. Some models are trained for one epoch to avoid overfitting. DomCLS is trained on the same data as DST models. We train the value tagger using Adam optimizer with a linear learning rate schedule that initiates at 2e-5. For other models, we use Adafactor optimizer with a linear learning rate schedule that initiates at 1e-3. To generate diverse rewrites $U^{conv}$ for our data synthesis, we use random sampling for the CoQR-R model. For CoQR-Zero and CoQR-SYN that generate self-contained rewrites $U^{sc}$, we use the beam search algorithm with a beam size of 5. For QADST and DST models, we use the greedy decoding strategy.

| | Training Data | DAC | SGD (19 domains) | | | | | MultiWOZ (5 domains) | | | |
|---|---|---|---|---|---|---|---|---|---|---|---|
| | | | RSA | | | | CDTA | RSA | | | CDTA |
| | | | 1 | 2 | 3 | 4 | | 1 | 2 | 3 | |
| **T5-DST** | SINGLE | ✗ | 94.1 | 44.9 | 23.5 | 13.2 | 2.7 | 85.9 | 44.4 | 22.1 | 1.6 |
| | | ✓ | 93.8 | 71.8 | 59.5 | 48.4 | 11.8 | 85.1 | 65.2 | 54.1 | 6.6 |
| | SINGLE+CONCAT | ✗ | 94.0 | 78.1 | 53.5 | 32.9 | 1.6 | 85.3 | 83.9 | 44.9 | 11.2 |
| | | ✓ | 93.9 | 76.9 | 74.7 | 50.9 | 1.8 | 85.3 | 83.6 | 68.4 | 11.1 |
| | SINGLE+SYN | ✗ | 94.0 | 81.8 | 54.3 | 33.1 | 25.8 | 85.1 | 82.4 | 50.7 | 20.9 |
| | | ✓ | 92.9 | 81.3 | 77.0 | 66.7 | **26.9** | 84.8 | 81.8 | 70.4 | **30.8** |
| **MinTL** | SINGLE | ✗ | 92.3 | 47.4 | 29.8 | 12.9 | 2.1 | 83.0 | 52.7 | 37.2 | 1.8 |
| | | ✓ | 93.5 | 64.3 | 56.2 | 37.0 | 0.8 | 83.9 | 59.3 | 44.4 | 1.9 |
| | SINGLE+CONCAT | ✗ | 92.8 | 70.0 | 67.8 | 54.3 | 1.1 | 84.2 | 81.5 | 64.0 | 5.8 |
| | | ✓ | 93.4 | 74.4 | 72.2 | 58.3 | 0.5 | 84.4 | 82.0 | 65.1 | 7.3 |
| | SINGLE+SYN | ✗ | 91.7 | 70.2 | 64.0 | 54.5 | 18.2 | 83.5 | 80.0 | 66.3 | 27.5 |
| | | ✓ | 92.9 | 78.0 | 73.4 | 62.9 | **21.8** | 83.6 | 80.4 | 66.3 | **29.1** |
| **SDP-DST** | SINGLE | ✗ | 93.6 | 59.0 | 44.7 | 47.7 | 3.8 | 86.2 | 58.0 | 39.8 | 1.4 |
| | | ✓ | 94.7 | 74.7 | 66.2 | 54.6 | 13.2 | 86.2 | 66.9 | 54.0 | 8.5 |
| | SINGLE+CONCAT | ✗ | 93.3 | 81.9 | 79.5 | 66.6 | 0.8 | 87.2 | 84.9 | 68.5 | 9.1 |
| | | ✓ | 95.0 | 78.3 | 77.5 | 58.7 | 1.7 | 86.6 | 84.3 | 68.1 | 8.3 |
| | SINGLE+SYN | ✗ | 92.6 | 83.4 | 80.1 | 74.6 | 18.1 | 86.9 | 83.8 | 72.4 | 26.8 |
| | | ✓ | 94.8 | 81.9 | 78.4 | 66.4 | **19.7** | 86.2 | 82.6 | 70.6 | **27.7** |

Table 6: DST performance on real multi-domain dialogs. We divide turns based on the number of domains in $B_t$ and evaluate RSA separately. When there are more domains in $B_t$ in test dialogs than training dialogs, the better result between models with/without DAC paradigm is in  gray . The best CDTA for each model is in **bold**.

| Training Data | SGD | | MultiWOZ | |
|---|---|---|---|---|
| | Single | Multi | Single | Multi |
| SINGLE | 98.7 | 85.3 | 99.8 | 82.8 |
| SINGLE+CONCAT | 98.6 | 90.7 | 99.8 | 94.8 |
| SINGLE+SYN | 98.5 | 90.9 | 99.7 | 94.4 |

Table 7: Domain classifier accuracy.

# 6 Experiments and Analysis

## 6.1 Main Experiment

We first examine the performance of the domain classifier, which is used by all models based on the DAC paradigm. As shown in Table 7, Dom-CLS accurately classifies single-domain dialogs and performs well on real multi-domain dialogs when trained on synthesized dialogs, allowing us to track the state of each domain separately.

Then we conduct experiments on three DST models to verify the effectiveness of using the DAC paradigm and our synthesized dialogs. Table 6 shows model performance on real multi-domain dialogs. We have the following findings:

- Models trained only on single-domain dialogs generalize poorly on dialogs involving more than one domain. Using the DAC paradigm or synthesizing multi-domain dialogs for training can significantly improve model generalization.
- For T5-DST and MinTL, applying the DAC paradigm often leads to better model generaliza-

tion on test dialogs involving more domains than training dialogs. SDP-DST already generalizes well when trained on synthesized multi-domain dialogs, and DAC leads to worse generalization. A possible reason is SDP-DST-DAC have not seen samples consisting of dialog and slots of inactive domains during training, thus may be confused by similar slots when predicted active domains are wrong. However, SDP-DST is infeasible when there are numerous domains due to low efficiency, and the DAC paradigm is a reasonable solution that substantially improves the training and inference efficiency.

- Compared with CONCAT, training models on our synthesized data SYN results in significantly better CDTA, which indicates SYN effectively enables models to capture value transfer between domains. Combining DAC and SYN leads to the best CDTA.

## 6.2 Generalization to Unseen Domain Combinations

We further investigate model generalization on unseen domain combinations. We first sample 15 domains from all 19 domains of the SGD dataset and then sample 10 domains from these 15 domains, creating $D_{:15}$ and $D_{:10}$, respectively. Then we train the model on the same amount of synthesized multi-

| T5-DST DAC | SYN Domains | RSA | | | CDTA | | |
|---|---|---|---|---|---|---|---|
| | | $D_{:10}$ | $D_{10:15}$ | $D_{15:19}$ | $D_{:10}$ | $D_{10:15}$ | $D_{15:19}$ |
| ✗ | $D_{:10}$ | 83.4 | 80.4 | 74.7 | 30.6 | 17.0 | 14.5 |
| | $D_{:15}$ | 83.1 | 85.1 | 76.3 | 44.2 | 23.9 | 19.9 |
| | $D_{:19}$ | 81.3 | 84.8 | 81.6 | 33.5 | 27.2 | 23.6 |
| ✓ | $D_{:10}$ | 87.0 | 84.1 | 82.1 | 38.0 | 27.7 | 21.0 |
| | $D_{:15}$ | 85.3 | 85.5 | 83.0 | 32.2 | 27.4 | 20.4 |
| | $D_{:19}$ | 87.5 | 85.6 | 83.7 | 38.4 | 27.2 | 24.3 |

Table 8: Model generalization on unseen domain combinations. The test multi-domain dialogs are divided into three disjoint sets based on their active domains: all in $D_{:10}$, at least one in $D_{10:15}$, and at least one in $D_{15:19}$. Results for dialogs with unseen domain combinations are in gray .

domain dialogs as SYN but involving domains in $D_{:10}/D_{:15}$ only, and evaluate model performance on dialogs containing unseen domain combinations (at least one domain in $D_{10:19}/D_{15:19}$). As shown in Table 8, the DAC paradigm makes T5-DST generalize better on unseen domain combinations (gray cells). Unlike T5-DST, synthesizing part of domain combinations is sufficient for T5-DST-DAC to perform well on all domain combinations.

### 6.3 CoQR On-the-Fly
The synthesized multi-domain dialogs can also be used to train a CoQR model to rewrite the utterance to express values explicitly, resolving cross-domain value transfer before DST. We compare the CoQR model trained on SYN with other CoQR models in Table 9. The DST model is T5-DST-DAC trained on SINGLE+CONCAT. We find that rewriting user utterances with CoQR-Zero or ChatGPT before DST severely sacrifices the overall performance for improving CDTA. In contrast, using CoQR-SYN trained on our synthesized rewrites can effectively address cross-domain value transfer while maintaining the overall performance. This is because CoQR-SYN is trained on in-domain data and therefore knows when and how to rewrite. We observe similar results on other DST models (Appendix A).

### 6.4 Ablation Study and Robustness Analysis
We conduct comprehensive experiments to identify key factors in data synthesis. Results are shown in Table 9. We have the following findings:
- Without utterance rewriting or value filter, the CDTA decreases significantly, indicating that high-quality rewrites teach models to capture the cross-domain value transfer behavior.
- Using 50% SYN data decrease the CDTA a little, while using twice SYN data does not guarantee performance improvement.

| T5-DST-DAC | SGD | | MultiWOZ | |
|---|---|---|---|---|
| | RSA | CDTA | RSA | CDTA |
| SINGLE+CONCAT | 82.5 | 1.8 | 83.2 | 11.1 |
| w/ CoQR-Zero | 75.8 | 16.6 | 78.9 | 11.4 |
| w/ CoQR-ChatGPT | 80.5 | 26.8 | 81.4 | 24.7 |
| w/ CoQR-SYN | 84.1 | 17.2 | 83.1 | 24.0 |
| SINGLE+SYN | 84.8 | 26.9 | 82.2 | 30.8 |
| w/o utterance rewrite | 82.8 | 4.4 | 82.7 | 20.2 |
| w/o filter | 84.4 | 21.1 | 81.5 | 27.6 |
| 0.5x SYN data | 85.0 | 24.0 | 81.7 | 28.4 |
| 2x SYN data | 83.8 | 26.5 | 81.2 | 29.8 |
| Cross-domain slot pairs | | | | |
| F1 > 0 | 84.6 | 26.2 | 81.9 | 29.1 |
| F1 > 0.1 | 84.8 | 26.9 | 82.2 | 30.8 |
| F1 > 0.3 | 85.2 | 24.0 | 81.8 | 30.6 |
| F1 > 0.5 | 85.1 | 26.2 | 81.5 | 26.9 |
| F1 > 0.8 | 83.9 | 18.1 | 81.7 | 21.5 |
| Mined from MULTI | 86.4 | 30.6 | 82.7 | 34.7 |
| 90%SINGLE+10%MULTI | 86.7 | 39.6 | 83.4 | 31.4 |

Table 9: CoQR on-the-fly experiments and ablation study. Results of using original SYN data are in gray .

- Performance does not change much when the F1 threshold of filtering slot pairs mined by QADST varies from 0 to 0.3. Higher F1 will filter out real related slots and lead to worse CDTA. The results of using slot pairs mined from real multi-domain dialogs for data synthesis suggest that improving slot relation mining can further enhance CDTA.
- Compared with replacing 10% single-domain dialogs with real multi-domain dialogs, using SYN for training achieves comparable results on MultiWOZ but worse results on SGD which contains much more domains. This indicates that the gap between real data and SYN is large on SGD, calling for better data synthesis methods.

## 7 Conclusion

In this paper, we investigate how to build multi-domain DST models using only single-domain dialogs. We propose a divide-and-conquer paradigm and a multi-domain dialog synthesis framework, enabling DST models to generalize better on unseen multi-domain dialogs and capture value transfer between domains. The DAC paradigm simplifies multi-domain DST to single-domain DST, improving model generalization and efficiency. Our synthesized multi-domain dialogs can be used to train DST models directly or train a CoQR model to rewrite utterances before DST. Both usages can significantly improve the domain relation modeling ability of DST models. Our methods can boost the development of multi-domain DST models by reducing the cost of collecting multi-domain dialogs.

# 8 Limitations

While our proposed methods have made initial progress in building multi-domain DST from single-domain dialogs, there is still much room for improvement. Currently, the DAC paradigm has two main drawbacks: (1) models could not access the dialog history of other domains and thus can only transfer values presented in the state, (2) the domain classifier and DST model are optimized independently, which may cause error propagation. Future works can retrieve relevant context dynamically and incorporate DST feedback in domain classification. The scope of multi-domain dialogs considered in this paper is rather simplistic, missing many kinds of domain interactions, such as domain composition (Andreas et al., 2020). The gap between synthesized multi-domain dialogs and real ones is still large, which can be improved through (1) utilizing commonsense knowledge in slot relation mining, (2) modeling diverse discourse structures, and (3) generating more reasonable rewrites with stronger CoQR models.

# Acknowledgements

This work was supported by the National Key Research and Development Program of China (No. 2021ZD0113304). This work was also supported by the National Science Foundation for Distinguished Young Scholars (with No. 62125604) and the NSFC projects (Key project with No. 61936010).

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

## A  Full Results of CoQR On-the-Fly

We show the full results of CoQR on-the-fly experiments in Table 10. From the results, we can observe that using the CoQR model to rewrite utterances before DST can improve CDTA. However, the CoQR model trained on CANARD (CoQR-Zero) severely decreases the overall performance (RSA). Using ChatGPT (`gpt-3.5-turbo-0301`) to rewrite utterances reduces the performance drop and achieves the best CDTA among CoQR models. We prompt ChatGPT with dialog history and ask it to rewrite the user utterance to resolve any anaphora or ellipsis. Several examples we used to prompt ChatGPT are demonstrated in Table 12. Using the CoQR model trained on SYN data (CoQR-SYN) can improve CDTA and maintain RSA. Directly training DST models on SYN data is better than using SYN data to train a CoQR model to rewrite utterances.

## B  Full Results of Slot Relation Mining

To evaluate the synthesized slot pairs, we extract "golden" slot pairs from real multi-domain dialogs, which may have noise. According to the state annotation, we extract the (source, target) slot pairs if P(target.value==source.value | source.value is not empty) larger than 10%.

We further report some statistics about the slot pairs in SYN data in Table 11. Increasing the F1 threshold will decrease the recall of golden slot pairs but improve precision. However, recalling more golden slot pairs seems more important than increasing precision. Comparing SGD and MultiWOZ, we find that our data synthesize framework can recall relatively more golden slot pairs on MultiWOZ, which may explain the performance gap between using SYN data and using real multi-domain data for training in Table 9. On SGD, even if we use the golden slot pairs for data synthesis, SYN data can not cover all golden slot pairs (only 75%) due to the value filtering process. On the contrary, we can almost synthesize all golden slot pairs (97.1%) on MultiWOZ.

## C  Prompt for ChatGPT to Generate Self-contained Rewrites

Table 12 shows the prompt we used for ChatGPT to rewrite the user utterance in the CoQR on-the-fly experiments. All user utterances in the dialog will be rewritten to resolve cross-domain value transfer.

| | | **SGD** | | **MultiWOZ** | |
|---|---|---|---|---|---|
| | | RSA | CDTA | RSA | CDTA |
| **T5-DST-DAC** | SINGLE+CONCAT | 82.5 | 1.8 | 83.2 | 11.1 |
| | w/ CoQR-Zero | 75.8 | 16.6 | 78.9 | 11.4 |
| | w/ CoQR-ChatGPT | 80.5 | 26.8 | 81.4 | 24.7 |
| | w/ CoQR-SYN | 84.1 | 17.2 | 83.1 | 24.0 |
| | SINGLE+SYN | 84.8 | 26.9 | 82.2 | 30.8 |
| **MinTL-DAC** | SINGLE+CONCAT | 80.9 | 0.5 | 81.8 | 7.3 |
| | w/ CoQR-Zero | 73.3 | 16.0 | 77.5 | 8.1 |
| | w/ CoQR-ChatGPT | 78.4 | 25.7 | 80.0 | 23.0 |
| | w/ CoQR-SYN | 82.7 | 16.9 | 81.5 | 21.6 |
| | SINGLE+SYN | 82.7 | 21.8 | 80.7 | 29.1 |
| **SDP-DST-DAC** | SINGLE+CONCAT | 84.0 | 1.7 | 84.1 | 8.3 |
| | w/ CoQR-Zero | 76.5 | 17.9 | 79.6 | 9.4 |
| | w/ CoQR-ChatGPT | 81.2 | 27.3 | 82.3 | 24.8 |
| | w/ CoQR-SYN | 85.6 | 17.8 | 83.8 | 22.6 |
| | SINGLE+SYN | 86.0 | 19.7 | 83.2 | 27.7 |

Table 10: CoQR on-the-fly experiments. Results of using original SYN data are in ` gray `.

| **T5-DST-DAC** | **SGD** | | | | | **MultiWOZ** | | | | |
|---|---|---|---|---|---|---|---|---|---|---|
| **SYN Slot Pairs** | RSA | CDTA | Ratio | Precision | Recall | RSA | CDTA | Ratio | Precision | Recall |
| F1 > 0 | 84.6 | 26.2 | 4.7 | 10.9 | 58.6 | 81.9 | 29.1 | 14.9 | 29.1 | 88.2 |
| F1 > 0.1 | 84.8 | **26.9** | 4.5 | 10.8 | 55.2 | 82.2 | **30.8** | 11.2 | 35.1 | 79.4 |
| F1 > 0.3 | 85.2 | 24.0 | 4.1 | 12.0 | 56.3 | 81.8 | 30.6 | 8.7 | 41.7 | 73.5 |
| F1 > 0.5 | 85.1 | 26.2 | 3.8 | 11.4 | 49.4 | 81.5 | 26.9 | 7.1 | 46.9 | 67.6 |
| F1 > 0.8 | 83.9 | 18.1 | 2.8 | 14.9 | 47.1 | 81.7 | 21.5 | 2.5 | 64.7 | 32.4 |
| Mined from MULTI | 86.4 | 30.6 | 0.7 | 100.0 | **75.9** | 82.7 | 34.7 | 4.8 | 100.0 | **97.1** |

Table 11: Training on SYN data constructed using different slot pairs. "Ratio" is the ratio of remaining slot pairs in SYN data compared to all possible slot pairs. "Precision" and "Recall" is calculated by comparing slot pairs in SYN data and slot pairs mined from MULTI. Significant performance drops are in ` gray `.

---

Replace the anaphora or ellipsis in this sentence with its real value in the context.
**Example 1:** Edit a sentence "I'd also like to find a restaurant there". The context is ["I'm bored. Can you find me a movie? I'm at Newark.", "There's 9 movies playing, including Breakthrough, Captain Marvel, and Dumbo."]. "there" is a pronoun refering a place, and it refers to "Newark" according to the context. So the edited sentence should be "I'd also like to find a restaurant at Newark".
**Example 2:**
Edit a sentence "no, but it should have free parking, please.". The context is ["Hi. I'm looking for a hotel in the east. The internet is not needed.", "There are no hotels that do not have internet, but 7 which do have it. Do you have a specific price range you'd prefer?"]. "it" refers to the hotel. So the edited sentence should be "no, but the hotel should have free parking, please.".
**Example 3:**
Edit a sentence "yeah book it for the same group of people please". The context is ["I'm looking a train that is leaving on Thursday but will arrive by 09:30 to Cambridge.", 'Okay, and where are you leaving from?', 'i will depart from birmingham new street', 'TR3736 arrives in Cambridge at 09:23.', "That would be great. Please book tickets for 7 people. I'll also need a reference number, if possible. ", 'Okay I booked it for you and your reference number is 0LYL0J3V.', "Thank you. I will also need a place to stay in the north. I'd like something that includes free parking.", 'Okay and what is your price range?', "It doesn't really matter as long as the parking is free. I don't even need internet.", 'when is your stay?', "Let's go with Thursday. ", 'Ok great. I highly suggest the Acorn Guest House. Would you like for me to book a room?']. The same group of people refers to 7 people in the context. So the edited sentence should be "yeah book it for the same group of people please".

If no anaphora or ellipsis exists, the edited utterance remains unchaneged.
The sentence you are supposed to edit is "{utterance}", and the context is {history}.
The edited sentence should be:

---

Table 12: The text input we used to prompt ChatGPT to rewrite the user utterance to resolve anaphora and ellipsis.