# OpenReview forum: "Building Multi-domain Dialog State Trackers from Single-domain Dialogs"
_EMNLP/2023/Conference — EMNLP 2023 Main_

### Official Review · Reviewer_9qPF · 2023-07-26

**Soundness:** 3

**Excitement:**

3: Ambivalent: It has merits (e.g., it reports state-of-the-art results, the idea is nice), but there are key weaknesses (e.g., it describes incremental work), and it can significantly benefit from another round of revision. However, I won't object to accepting it if my co-reviewers champion it.

**Paper Topic And Main Contributions:**

This paper proposes a divide-and-conquer paradigm for dialog state tracing (DST). The method divides dialogs according to domains and mixt them to synthesis multi-domain data, providing a data augmentation method to improve cross-domain DST. Experiment results show this method boosts multi-domain DST performance using single-domain data.

**Questions For The Authors:**

See above.

**Reasons To Accept:**

1. The result is positive and also helpful for reducing the cost of data collection for multi-domain DST.
2. The method is data-centric and thus generic for various models. Potentially, this method can also benefit other domains relating to compositional generalization.
2. The writing is clear and easy to follow.

**Reasons To Reject:**

1. The author should provide discussions to support their motivation to work on DST, a component that serves task-oriented dialog systems (TOD). In particular, the authors should compare TOD with general-purpose chatbots like ChatGPT.
2. This paper only investigates generative DST models with only a fixed scale T5 as the backbone. Whether the improvement in compositional generalization remains considerable with non-generative methods and larger models remains unclear.

**Reproducibility:**

4: Could mostly reproduce the results, but there may be some variation because of sample variance or minor variations in their interpretation of the protocol or method.

**Reviewer Confidence:**

4: Quite sure. I tried to check the important points carefully. It's unlikely, though conceivable, that I missed something that should affect my ratings.

---

> ### Author Rebuttal · Authors · 2023-08-25
>
> Thanks for your constructive comments and feedback. We appreciate that you found (1) our method is effective and can reduce the cost of multi-domain data collection; (2) our method is generic and potentially can be used in other scenarios relating to compositional generalization; (3) the writing is clear and easy to follow.
>
> We have carefully considered your comments and would like to respond point-by-point:
> > 1. The author should provide discussions to support their motivation to work on DST, a component that serves task-oriented dialog systems (TOD). In particular, the authors should compare TOD with general-purpose chatbots like ChatGPT.
>
> Our research interest is task-oriented dialog (TOD), and we think the most challenging and valuable problem of multi-domain dialogs compared with single-domain dialogs is the recognition of cross-domain value transfer when domain relations are undefined, so we investigate multi-domain DST that needs to model value transfer between domains. We are unsure about the meaning of "compare TOD with general-purpose chatbots like ChatGPT":
> 1. Did you mean using ChatGPT for TOD? Recent studies[1,2,3] show that ChatGPT achieves SOTA zero-shot DST results on TOD tasks but still largely underperforms fine-tuned models. So in this paper, we did not use ChatGPT for DST.
> 2. Or did you mean we should discuss the difference between TOD and general-purpose chatbots? We do have a few opinions about the difference and what LLMs like ChatGPT need so as to serve as TOD systems. However, we do not include these ideas since they are not strongly related to the focus of this paper.
>
> We would love to discuss with you to figure out what you think exactly.
>
> > 2. This paper only investigates generative DST models with only a fixed scale T5 as the backbone. Whether the improvement in compositional generalization remains considerable with non-generative methods and larger models remains unclear.
>
> Thanks for your suggestions! As we stated in Section 3.2, we use open-vocabulary generative DST models because they are flexible to generate any value and require minimal annotation. Classification-based models need full value lists and can not predict values out of the lists, while extraction-based models need value span annotations and can not predict values that are not presented in the dialog context. Besides, most Large Language Models (LLMs) are generative models. So we conducted experiments using generative DST considering community interest. Although we do not try non-generative methods, we explore three different kinds of generative models to show the intrinsic difficulty in modeling cross-domain value transfer (often through anaphora) without real multi-domain data. In fact, we think most current multi-domain DST models would get into trouble with this problem, and our Divide-and-Conquer paradigm and data synthesis framework would help.
>
> We use T5-large because it's the largest model we can afford to conduct comprehensive experiments. For LLMs that show emergent abilities[4], we do not have sufficient computation resources to fine-tune them. As stated above, without fine-tuning, ChatGPT could not give satisfactory DST performance[1,2,3].
>
> We do not intend to deny the limitations of our work. We just want to explain the reasons for our choices in detail, in response to your valuable feedback. Thanks for your suggestions. We will verify the effectiveness of our method on non-generative methods and larger models in the future. We kindly ask you to reconsider our paper.
>
> ---
>
> [1] ChatGPT for Zero-shot Dialogue State Tracking: A Solution or an Opportunity? (Heck et al., ACL 2023)
>
> [2] Are LLMs All You Need for Task-Oriented Dialogue? (Hudeček and Dušek, SIGDial 2023)
>
> [3] A Preliminary Evaluation of ChatGPT for Zero-shot Dialogue Understanding (Pan et al., arxiv 2023)
>
> [4] Emergent Abilities of Large Language Models (Wei et al., TMLR 2022)

---

### Official Review · Reviewer_DJWY · 2023-08-02

**Soundness:** 4

**Excitement:**

3: Ambivalent: It has merits (e.g., it reports state-of-the-art results, the idea is nice), but there are key weaknesses (e.g., it describes incremental work), and it can significantly benefit from another round of revision. However, I won't object to accepting it if my co-reviewers champion it.

**Paper Topic And Main Contributions:**

This paper discusses building multi-domain Dialog State Tracking (DST) models using single-domain dialogs. The main problem the paper addresses is the difficulty and expenses involved in manual domain relation definition and multi-domain dialog data collection. Hence, the authors proposed a divide-and-conquer (DAC) DST paradigm and a multi-domain dialog synthesis framework. The divided and conquer approach generalizes better on dialogs involving unseen domain combinations while the multi-domain dialog synthesis framework simulates domain relationships by merging single-domain dialogs and modifying them. The experiments conducted on two datasets demonstrate the efficacy of these proposed strategies.

**Questions For The Authors:**

- In section 5.2.1, the authors said “For a dialog of domain d, the model is trained to predict the values of slots in d. When performing cross-domain inference, the model predicts the values of target slots in other domains.” This is a bit unclear for me. How to get all candidate slots and values in cross-domain inference? Do you iterate over all of the domains except d to get all candidate slots, or train a dedicated model for each domain?
- Theoretically, the information of belief states of upstream domains is enough if the correct belief states are sent to the downstream domains. However, the errors in the belief states may also propagate to downstream domains. How to avoid this in the divide-and-conquer DST paradigm?
- How to make sure the slot pairs that could potentially take the same value in a specific dialog context really make sense? Is there any method to validate if the synthetic dialogs are reasonable?

**Reasons To Accept:**

- The question raised in this paper is interesting and valuable in real-world scenarios: How to train an effective multi-domain DST model when we only have single-domain dialogs?
- This paper proposes a feasible solution to tackle this problem. The paper is well-written and easy to follow.
- The paradigm proposed in this paper can be generalized to a series of DST models.
- The experimental results demonstrate the effectiveness of the proposed solution.

**Reasons To Reject:**

- The divide-and-conquer DST paradigm is to model multi-domain DST in a pipelined single-domain manner with the dialog states from the upstream domains. The pipeline solutions will suffer from some intrinsic weaknesses such as error propagation. In contrast, previous multi-domain DST work can be seen as an end-to-end solution since they can always generate or update all domains and slots in one turn.
- Considering the data is synthetic rather than natural, it may introduce errors in the multi-domain dialog synthesis framework. Although the author proposes a filtering process by a value tagger, it cannot make sure that the automatically mined slot pairs indeed make sense.

**Reproducibility:**

4: Could mostly reproduce the results, but there may be some variation because of sample variance or minor variations in their interpretation of the protocol or method.

**Reviewer Confidence:**

4: Quite sure. I tried to check the important points carefully. It's unlikely, though conceivable, that I missed something that should affect my ratings.

---

> ### Author Rebuttal · Authors · 2023-08-27
>
> Thanks for your constructive comments and feedback. We appreciate that you found (1) the problem we investigate is interesting, and our solution is feasible and effective; (2) our proposed paradigm can be generalized to a series of DST models; (3) the paper is well-written and easy to follow.
>
> We have carefully considered your comments and would like to respond point-by-point:
>
> > The divide-and-conquer DST paradigm is to model multi-domain DST in a pipelined single-domain manner with the dialog states from the upstream domains. The pipeline solutions will suffer from some intrinsic weaknesses such as error propagation. In contrast, previous multi-domain DST work can be seen as an end-to-end solution since they can always generate or update all domains and slots in one turn.
>
> We agree this is a limitation of the current divide-and-conquer (DAC) paradigm. One of the reasons that we choose to optimize the domain classifier and DST models independently is to make a fair comparison between different DST models with the same domain classifier. Compared with DST models that always predict all domains and slots, models applying the DAC paradigm are more efficient and generalize better on multi-domain dialogs that have unseen domain combinations (Table 8) and more domains than training (Table 6). Therefore, although the DAC paradigm has some weaknesses, we think it is more suitable for the scenario where the distribution of real multi-domain dialogs is unknown.
>
> > Considering the data is synthetic rather than natural, it may introduce errors in the multi-domain dialog synthesis framework. Although the author proposes a filtering process by a value tagger, it cannot make sure that the automatically mined slot pairs indeed make sense.
>
> We agree. Actually, we do observe some erroneous slot pairs. We think the slot mining process can be further improved by using large language models that possess commonsense knowledge. We do not use ChatGPT to do this because slot pair mining is the first step, and using ChatGPT (not open-source, not stable) will greatly affect the reproducibility of the work. At the time we conduct experiments, open-source LLMs like LLaMa are not available. In the future, we will use these open-source LLMs to mine slot pairs that could potentially transfer value.
>
> > In section 5.2.1, the authors said “For a dialog of domain d, the model is trained to predict the values of slots in d. When performing cross-domain inference, the model predicts the values of target slots in other domains.” This is a bit unclear for me. How to get all candidate slots and values in cross-domain inference? Do you iterate over all of the domains except d to get all candidate slots, or train a dedicated model for each domain?
>
> We train a single model on all single-domain dialogs. For each single-domain dialog belonging to domain $d$, the model is only optimized to predict the slots of $d$. During cross-domain inference, for a single-domain dialog, we iterate over all of the domains except $d$ to get all candidate slots.
>
> > Theoretically, the information of belief states of upstream domains is enough if the correct belief states are sent to the downstream domains. However, the errors in the belief states may also propagate to downstream domains. How to avoid this in the divide-and-conquer DST paradigm?
>
> This question enlightens us! Actually, this could be one of the reasons that SDP-DST (predicts all slots at each turn with all dialog history as input) performs better than SDP-DST with DAC (predicts slots of the active domain at each turn with only current domain dialog as input) in Table 6. To alleviate this problem, the DAC paradigm may also take relevant dialog context as input through retrieval. For example, when synthesizing the turn $t$ that has target slots for cross-domain value transfer, we can also create retrieval samples for retrieving turns $(t^\prime-w, t^\prime+w)$, where $t^\prime$ is the turn that source slots appear, and $w$ is the context window size. In this way, the DST models can directly refer to the relevant context instead of the model-predicted state when solving cross-domain value transfer.
>
> > How to make sure the slot pairs that could potentially take the same value in a specific dialog context really make sense? Is there any method to validate if the synthetic dialogs are reasonable?
>
> As mentioned above, we plan to use LLMs that possess commonsense knowledge to identify slot pairs that could potentially take the same value in a specific dialog context. We think it is also a plausible way to validate if the synthetic dialogs are reasonable.
>
> Thank you for your enlightening comments! We will continue refining our work according to your advice.

---

### Official Review · Reviewer_NSN3 · 2023-08-04

**Typos Grammar Style And Presentation Improvements:** Wrong template is used
**Soundness:** 3

**Excitement:**

4: Strong: This paper deepens the understanding of some phenomenon or lowers the barriers to an existing research direction.

**Paper Topic And Main Contributions:**

This paper focuses on improving multi-domain DST with single-domain data. It proposes a new divide-and-conquer approach and data synthesis method. The method reduces the need for expensive multi-domain dialogues labeling. The results demonstrate the cross-domain generalization.

**Reasons To Accept:**

1. The paper proposes two novel methods, including divide-and-conquer paradigm and data synthesis framework, to address the key challenges of building multi-domain models without multi-domain data.
2. Comprehensive experiments demonstrate the effectiveness of the proposed methods on two benchmark datasets using three representative dialog state tracking models. Results show significant improvements in model generalization and capturing cross-domain value transfer.

**Reasons To Reject:**

1. Error propagation could be an issue as the domain classifier and state tracker are optimized independently in the divide-and-conquer paradigm.
2. The association between DAC method and multi-domain dialog synthesis framework is relatively weak, insufficient to constitute a cohesive unified framework.

**Reproducibility:**

4: Could mostly reproduce the results, but there may be some variation because of sample variance or minor variations in their interpretation of the protocol or method.

**Reviewer Confidence:**

3: Pretty sure, but there's a chance I missed something. Although I have a good feel for this area in general, I did not carefully check the paper's details, e.g., the math, experimental design, or novelty.

---

> ### Author Rebuttal · Authors · 2023-08-27
>
> Thanks for your constructive comments and feedback. We appreciate that you found our methods are effective and the improvements are significant. We have carefully considered your comments and would like to respond point-by-point:
>
> > Error propagation could be an issue as the domain classifier and state tracker are optimized independently in the divide-and-conquer paradigm.
>
> We agree this is a limitation of the current divide-and-conquer (DAC) paradigm. One of the reasons that we choose to optimize the domain classifier and DST models independently is to make a fair comparison between different DST models with the same domain classifier. Future works can improve the DAC paradigm by correcting the domain classifier's prediction with feedback from the DST model.
>
> > The association between DAC method and multi-domain dialog synthesis framework is relatively weak, insufficient to constitute a cohesive unified framework.
>
> We consider building multi-domain DST using single-domain data in two perspectives: model and data. Investigating only one of them could not show its necessity since one could argue that the problem can be solved by improving the other (better model/data synthesis method). Indeed, our experiments show that both the DAC paradigm and the multi-domain data synthesis framework are necessary and complement each other.
>
> > Wrong template is used: "Anonymous ACL submission"
>
> Thanks for your kind reminder! We will modify this.

---

### Meta-Review · Area_Chair_oiYt · 2023-09-18

**Recommendation:** 4

**Metareview:**

This work proposes constructing multi-domain Dialog State Tracking (DST) models using a divide-and-conquer (DAC) DST paradigm along with multi-domain dialogue synthesis from existing single-domain dialogues.

Contributions:
 - The paper proposes two novel methods, including divide-and-conquer paradigm and data synthesis framework, to address the key challenges of building multi-domain models without multi-domain data.
 - The method is data-centric and thus generic for various models.
 - Experiments demonstrate the effectiveness of the proposed methods.

---

### Decision · Program_Chairs · 2023-10-07

**Decision:**

Accept-Main

**Comment:**

This work proposes constructing multi-domain Dialog State Tracking (DST) models using a divide-and-conquer (DAC) DST paradigm along with multi-domain dialogue synthesis from existing single-domain dialogues.

Contributions:
 - The paper proposes two novel methods, including divide-and-conquer paradigm and data synthesis framework, to address the key challenges of building multi-domain models without multi-domain data.
 - The method is data-centric and thus generic for various models.
 - Experiments demonstrate the effectiveness of the proposed methods.